# Divergent effects of azithromycin on purple corn (*Zea mays L*.) cultivation: Impact on biomass and antioxidant compounds

**Yoselin Mamani Ramos**[1,2], **Nils Leander Huamán Castilla**[3,4], **Elvis Jack Colque Ayma**[2], **Noemi Mamani Condori**[1,2], **Clara Nely Campos Quiróz**[2], **Franz Zirena Vilca**[1,2]*

1 Escuela Profesional de Ingeniería Ambiental de la Universidad Nacional de Moquegua, Urb Ciudad Jardín-Pacocha-Ilo, Perú, 2 Laboratorio de Contaminantes Orgánicos y Ambiente del IINDEP de la Universidad Nacional de Moquegua, Urb Ciudad Jardín-Pacocha-Ilo, Perú, 3 Escuela Profesional de Ingeniería Agroindustrial, Universidad Nacional de Moquegua, Moquegua, Perú, 4 Laboratorio de Tecnologías Sustentables para la Extracción de Compuestos de Alto Valor, Instituto de Investigación para el Desarrollo del Perú, Universidad Nacional de Moquegua, Moquegua, Perú

* fzirenav@unam.edu.pe

**Data Availability Statement:** All data are in the manuscript and/or supporting information files.

## Abstract

The present study assessed the impact of using irrigation water contaminated with Azithromycin (AZM) residues on the biomass and antioxidant compounds of purple corn; for this purpose, the plants were cultivated under ambient conditions, and the substrate used consisted of soil free from AZM residues, mixed with compost in a ratio of 1:1 (v/v). The experiment was completely randomized with four replications, with treatments of 0, 1, 10, and 100 μg/L of AZM. The results indicate that the presence of AZM in irrigation water at doses of 1 and 10 μg/L increases the weight of dry aboveground biomass, while at an amount of 100 μg/L, it decreases. Likewise, this study reveals that by increasing the concentration of AZM from 1 to 10 μg/L, total polyphenols and monomeric anthocyanins double, in contrast, with an increase to 100 μg/L, these decrease by 44 and 53%, respectively. It has been demonstrated that purple corn exposed to the antibiotic AZM at low doses has a notable antioxidant function in terms of DPPH and ORAC. The content of flavonols, phenolic acids, and flavanols increases by 57, 28, and 83%, respectively, when the AZM concentration is from 1 to 10 μg/L. However, with an increase to 100 μg/L, these compounds decrease by 17, 40, and 42%, respectively. On the other hand, stem length, root length, and dry weight of root biomass are not significantly affected by the presence of AZM in irrigation water.

## Introduction

Purple corn (*Zea Mays L*.) is a distinct variety of maize emblematic of Peru, whose production reached 24.5 thousand tons in 2020, as the Ministry of Agrarian Development and Irrigation reported. This product has gained considerable attention recently due to its rich polyphenol and anthocyanin content and potential health benefits [1]. In particular, the corn crown has high concentrations of polyphenols (225 mg GAE/100 g dw) and anthocyanins (2600–3800

**Funding:** The Universidad Nacional de Moquegua funded this study through the following resolutions: Resolución N° 0310–2020—UNAM, Resolución de Comisión Organizadora N° 059–2021—UNAM, and Resolución de Comisión Organizadora N° 727–2022—UNAM. The funders had no role in study design, data collection and analysis, decision to publish, or preparation of the manuscript.

**Competing interests:** The authors have declared that no competing interests exist.

mg/100 g dw) [2, 3], which have demonstrated their effectiveness in the treatment of diseases related to oxidative stress [4, 5]. For example, phenolic acids, flavonols, and stilbenes are polyphenols used to treat conditions like hypertension, intestinal inflammation, and cancer [6, 7]. On the other hand, anthocyanins like cyanidin and delphinidin can be used to treat type 2 diabetes [8]. Thus, this corn variety is a bioactive compound source and a valuable ingredient in the food and beverage industry.

Polyphenols have phenolic and hydroxyl groups in their chemical structure, and their molecular weights can vary between 160 and 30,000 Da [9, 10]. These compounds are synthesized through the phenylpropanoid metabolic pathway as a protective response to stressors like temperature fluctuations, humidity levels, exposure to ultraviolet radiation, and exposure to chemical compounds such as pesticides and antibiotics [11–13]. In this sense, antibiotics and pharmaceutical products can disrupt plant development and impact the synthesis of polyphenols.

Antibiotics and pharmaceuticals are used to improve public health. However, these substances are not completely metabolized, leading to the presence of their residues in surface and wastewater [14, 15]. Antibiotic residues in aquatic environments are a cause for concern, as they promote the transfer of antibiotic resistance genes (ARG) among different bacteria [16]. One potential source of these residues is wastewater treatment plants, whose insufficient infrastructure contributes to the spread of pollutants in water bodies [17]. For example [18], has identified the presence of elevated concentrations of various antibiotic residues in surface waters, such as sulfamethoxazole, trimethoprim, azithromycin (AZM), clarithromycin, and ciprofloxacin (CPF). Similarly [19], has found antibiotic residues in drinking water, with azithromycin being the compound with the highest detection frequency (79.71–100%).

Antibiotics can be adsorbed and distributed among plant tissues like roots, stems, leaves, and fruits [20, 21]. For example, when AZM (23.6 ng/L) and CPF (15.6 ng/L) are present in irrigation water, plants can absorb between 5 and 15% of these compounds [22]. Although the presence of antibiotics in the plant could affect its normal development, these compounds have the potential to stimulate an increased production of polyphenols. For example [11], demonstrated in quinoa grains that controlled exposure to CPF residues in irrigation water (100 μg/L) allowed an increase in the polyphenol content by 52% compared to the control treatment. In contrast, under these considerations, morphological characteristics such as roots, leaves, panicles, and grain were not affected; however, different plants possess variable vulnerability to other antibiotics, even inducing a hormetic response. This study explores the influence of AZM residues in irrigation water on the impact of biomass and antioxidant compounds in purple corn.

## Materials and methods

The experiment was conducted under ambient conditions, using polyethylene pots with dimensions of 24 X 31.5 X 28 cm for width, height, and base, respectively, filled with 18 kg of the substrate (soil and compost) up to a height of 27 cm. Subsequently, after the plants completed their physiological cycle, the length and dry weight of the root biomass, stem length, dry weight of the aboveground biomass, and dry weight of the cob were analyzed. The total phenolic compounds and anthocyanin concentrations in the cob were also examined.

The planting was carried out according to the procedure proposed by [23] with minimal adjustments based on the observations of [24]. The seeds used in the current study were of the INIA 601 variety, provided by the National Institute of Agricultural Innovation–INIA. In the first phase, the seeds underwent surface disinfection using a 2.5% sodium hypochlorite solution with an exposure time of 5 minutes. Subsequently, rinses with distilled water were carried

out, and the seeds were placed in 18-litre pots containing a substrate mixture of soil and compost in a 1:1 ratio (v/v). Initially, each bank accommodated four sources, and after the emergence of seedlings, the most vigorous plant in each pot was selected, resulting in a single plant per container. The cultivation occurred under ambient conditions, and the plants received initial irrigation with distilled water.

The AZM stock solution was prepared in distilled water using analytical-grade AZM. This stock solution was subsequently used to create different test concentrations. The plants were irrigated with concentrations of 0 (control with distilled water) and 1, 10, and 100 µg/L of AZM, taking as a reference a previously documented range of occurrence of this antibiotic in surface waters [18]. The irrigation of the plants was carried out two or three times per week, using between 1 and 2 liters of water on each occasion, aiming to maintain a substrate field capacity of 70%. Sixteen pots were arranged and distributed across four treatments, including a control (without antibiotic) and concentrations of 1, 10, and 100 µg/L of AZM, with four replications for each treatment.

## Evaluation of the dry biomass of the plant

The method was proposed by [24] to assess the height and weight of the dry aboveground biomass, root length, and dry weight of the root biomass. At the physiological maturity phase, known as stage R6, the plants/treatments were harvested. They were laid out on a plastic surface to measure root length and stem length, the latter from the base of the stem to the top end of the corn tassel.

Subsequently, each plant was carefully divided into leaves, stems, roots, tassels, and cobs on a plastic tray. Each component was meticulously cleaned to remove impurities. Afterward, the roots, stems, leaves, and tassels were dried in an oven at 60˚C for 48 hours and then weighed on an analytical balance.

Only the cobs were removed and dried at 35˚C in the oven. This process was conducted at that temperature to allow for subsequent analysis of phenolic compounds and anthocyanins. The decision to analyze only the phenolic compounds in the cobs is justified by the concentration and differential distribution of these compounds in the plant and by considerations related to the study's objectives.

## Extraction of phenolic compounds

The cob was ground to obtain a fine powder (approximately 500 µm), and subsequently, the sample's moisture content was determined to extract phenolic compounds.

The extraction of phenolic compounds was carried out using the method reported by [25] with slight modifications based on what was written by [26]. To extract phenolic compounds, 1g of dry powder sample was mixed with 20 mL of 60% acetone (acetone/water 60:40, v/v). The mixture was then stirred (350 rpm for 30 minutes at 35˚C). Subsequently, the homogenized mixture was centrifuged at 4000 rpm for 5 minutes, and the supernatant was recovered. A second centrifugation at 4000 rpm for 5 minutes was performed on the residue, and the second supernatant was recovered. Both supernatants were combined and concentrated under vacuum at 40˚C until dryness. For this, the vacuum pressure was adjusted to 7.3 kPa, facilitating the evaporation of acetone and pure water at that temperature. The phenolic compounds were then reconstituted in 10 mL of 60% acetone. A first volume of 5mL was added to the rotary evaporator flask to recover the adhered mass. Then, the mixture was added to a vial, and 60% (v/v) acetone was added until a total volume of 10 mL was made up. The final aqueous extract obtained was stored in amber containers and kept at a temperature of -20˚C until further analysis.

## Quantification of total phenolic compounds

The concentration of total phenolic compounds in the crude extract was determined using the method proposed by [27]. In brief, 0.5 mL of corn extract was diluted with 3.75 mL of distilled water in a flask. 0.25 mL of Folin-Ciocalteu reagent (1 N) was added to each sample. After 5 minutes of reaction, 0.5 mL of a 10% aqueous solution of sodium carbonate ($Na_2CO_3$) was added. The mixture was then allowed to stand at room temperature for one hour until the characteristic blue color developed. Absorbance was measured at 765 nm against a blank containing distilled water instead of the sample using a spectrophotometer. The total phenolic compound content was determined through a calibration curve prepared with gallic acid at different concentrations ranging from 10 to 90 mg/L, with a high coefficient of determination ($R2$) of 0.9988, where the calibration curve is $y = 0.009x+0.00174$ The results were expressed as milligrams of gallic acid equivalents (GAE) per gram of the sample's dry weight (DW).

$$PFT\left(\frac{mg\ Gallic\ acid}{litre}\right) = \frac{(Abs\ sample - B)}{A} \times FD \tag{1}$$

Where Abs. The sample is the absorbance of the sample, A Slope, B Intercept, and FD dilution factor. Later, this value was corrected to mgGAE/gdw, considering values obtained during the moisture determination.

## Total Anthocyanin Content (TAC)

The total anthocyanin content was determined using the differential pH method [28], with some modifications reported [29]. The entire anthocyanin content was also determined from the crude extract. Before TAC determination, two solutions were prepared, one for pH 1.0 using potassium chloride buffer (0.025 M, 0.0465 g of KCl in 25 mL of distilled water, acidified with HCl), and the other for pH 4.5 using sodium acetate buffer (0.4 M, 1.3608 g of $CH_3CO_2Na$ in 25 mL of distilled water, acidified with HCl). Subsequently, the protocol for total anthocyanins was followed, which involved 20 μL of extract in 80 μL of pH buffer and 20 μL of section in 80 μL of pH 4.5 buffer. Each sample's absorbance was measured at 520 nm and 700 nm. TAC was calculated using the following equation and is expressed as milligrams of cyanidin-3-glucoside (C3G) per gram of dry weight (dw) of the sample

$$TAC\left(\frac{mg\ EC3G}{L}\right) = \frac{A \times MW \times FD \times 1000}{eL} \times \frac{V}{g\ fw} \times \frac{g\ fw}{g\ dw} \tag{2}$$

Where: TAC Is the concentration of total anthocyanins (mg EC3G/g bs); A It is the absorbance and is calculated as A = (A520 –A700) pH 1.0 - (A520 –A700) pH 4.5); e is the molar absorptivity of cyanidin-3-glucoside (26900 L/ (cm * mol)); L is the cell path length (1 cm); MW is the molecular weight of cyanidin-3-glucoside (449.2 g/mol); FD is the dilution factor. Subsequently, this value was corrected to mgEC3G/gdw, considering values obtained during the moisture determination: g fw (grams wet weight) and g dw (grams dry weight).

## Antioxidant capacity by DPPH

The antioxidant capacity of the extracts was evaluated using the DPPH (2,2-diphenyl-1-picrylhydrazyl) radical scavenging method [11, 30]. Firstly, the crude extract was diluted considering the following dilution factors: 20, 40, and 60 for the control group, while for the groups treated with 1, 10, and 100 μg/L of AZM, the dilutions were 40, 60, and 80. These dilutions were designed explicitly for our extracts and were used in this analysis. The test involved mixing 0.1 mL of extract with 3.9 ml of a DPPH solution (0.1 mM). The resulting combination was left at

room temperature in the dark for 30 minutes. After this period, the reduction of DPPH was measured by absorption at 517 nm using a UV spectrometer (UV 1240, Shimadzu, Kyoto, Japan). The determination of IC50, representing the concentration of extract needed to inhibit the absorption of DPPH radicals by 50%, was carried out based on these data. It is important to note that this parameter indicates the antioxidant efficacy of the extracts, providing valuable information about their ability to neutralize free radicals.

## Antioxidant capacity by Oxygen Radical Absorbance Capacity (ORAC)

The antioxidant activity of the extracts was determined according to the methodology proposed by [11, 31]. The ORAC analyses were performed on a 96-well microplate fluorometer (Ascent FL Fluoroscan, Labsystem, Finland). To generate peroxyl radicals, 2,2'-azobis(2-amidinopropane) dihydrochloride (AAPH) at a concentration of 153 mM was used. Trolox (0.01 M) was employed as the standard, and fluorescein (55 mM) served as the fluorescent probe, using approximately 25 µL of phosphate buffer (75 mM) at pH 7.4 as the blank. Then, Trolox standards were prepared at 8, 16, 24, 32, and 40 µM concentrations. Additionally, the samples were diluted in phosphate-buffered saline (PBS) solution at pH 7.4. These dilutions had a dilution factor of 8000 in the control group and the treatments with 1 and 100 µg/L of AZM, while in the treatment with 10 µg/L of AZM, the dilution factor was 13000. Subsequently, Trolox standard or the diluted sample in phosphate-buffered saline (PBS) at pH 7.4 was mixed with 250 µL of fluorescein, followed by a 10-minute incubation at 37°C. An automatic injection of 25 µL of AAPH solution (153 mM) was added to all microplates, and fluorescence was measured every minute for 50 minutes. The final ORAC values were calculated from the area under the curves, expressed as micromoles of Trolox equivalents per gram of dry weight (µmol TE/g dw). This quantitative method provides information about the antioxidant capacity of the extracts, evaluating their effectiveness in neutralizing peroxyl radicals.

## Detection and quantification of polyphenols

A modified methodology by [11] for detecting and quantifying the polyphenol profile. For the detection and quantification of the polyphenol profile, a modified solid-phase extraction (SPE) process was used, employing SiliaPrep C-18 cartridges (17%) 500 mg 3 mL, 60 A; previously conditioned with 10 mL of MeOH, 10 mL of ultra-pure water, and then the 8 mL sample; later, it was diluted with 10 mL of MeOH. Subsequently, it was filtered through a syringe filter with a pore size of 0.22 µm and finally placed in an ultra-high-resolution chromatography (UHPLC), Agilent 1290 II, USA, which is coupled with a Diode Array Detector (DAD). In addition, a Poroshell EC—C18 column (2.1 x 150 mm × 1.9 µm) was used. The chromatographic equipment was set at 30°C, a flow rate of 0.3 mL min-1, and an injection volume of 1 µL. However, the mobile phase preparation for separation was done using mobile phases such as A (ultra-pure water and 0.1% formic acid) and B (acetonitrile and 0.1% formic acid), which were in gradient mode. The gradient was programmed as follows: 95% A– 5% B for the first 15 min, followed by 60% A– 40% B for the next 18 min, and then a return to 95% A– 5% B over the next 20 min, with a post time of one min. The polyphenol standards considered were those established in Table 1. These standards were mixed and diluted at concentrations ranging from 0.3 to 10 µg/mL for the calibration curves. Additionally, the analyses were performed in triplicate.

## Data analysis

The statistical analysis used R Studio version 4.2.1 for the Windows system. The data underwent one-way analysis of variance (ANOVA) after verifying that they met the assumptions of

**Table 1. Stem length and dry aboveground biomass weight according to the different concentrations of AZM.**

| concentrations of AZM µg/L | Stem weight (grams) | | Leaf weight (grams) | | Stem length (centimeter) | | Tassel weight | |
|---|---|---|---|---|---|---|---|---|
| | $\bar{X}$ | SD | $\bar{X}$ | SD | $\bar{X}$ | SD | $\bar{X}$ | SD |
| 0 | 62.64[ab] | ±3.35 | 36.15[b] | ±4.76 | 209[a] | ±20.42 | 7.2[a] | ±0.76 |
| 1 | 71.7[a] | ±8.21 | 50.31[a] | ±7.18 | 199[a] | ±21.74 | 4.8[b] | ±0.52 |
| 10 | 67.16[ab] | ±8.66 | 45.32[ab] | ±7.23 | 215[a] | ±21 | 4.2[bc] | ±1.22 |
| 100 | 53.18[b] | ±6.79 | 37.99[ab] | ±5.61 | 200[a] | ±7.54 | 3.38[c] | ±0.54 |

$\bar{X}$: Represents the average. SD: Represents the standard deviation (n = 3).

normality through the Shapiro-Wilk test and homogeneity of variances through Levene's test. Subsequently, a post hoc test was performed, with a significance level (alpha) of 0.05, to determine the presence of significant differences in the data.

## Results and discussion

### The effect of azithromycin on stem length and dry aboveground biomass

The results obtained regarding stem length and dry aboveground biomass (tassel weight, stem weight, and leaf weight) of purple corn (*Zea mays* L.) are presented in Table 1.

### Stem length

These findings reveal that *Zea mays* L. was not significantly affected by AZM in plant growth when applied at 1, 10, and 100 µg/L of AZM throughout its vegetative period (Table 1). Additionally, a slight increase of 3% in stem length was observed at a dose of 10 µg/L of AZM compared to the control group (Table 1). However, this positive effect disappeared when the concentration was raised to 100 µg/L of AZM, resulting in a 7% decrease in stem length compared to the 10 µg/L AZM dose, falling below the level of the control group. Therefore, these results suggest that purple corn appears resistant to AZM at low or moderate amounts, but high doses may significantly negatively impact its growth.

Stem length was not significantly affected despite existing scientific literature reporting adverse effects of antibiotic residues on various crops' sizes [32, 33]. Other response mechanisms of this particular species, such as secondary metabolites, could explain this apparent contradiction. Given that this crop is known for its high content of phenolic compounds, primarily flavonoids, and among them, anthocyanins are especially relevant [34]. These natural compounds are fundamental to the plant's defense [35]. Plants often use the apoplast and other specialized organelles, such as the vacuole, to store these compounds produced through the shikimic acid pathway to respond to external threats [36]. These phenolic compounds actively work to counteract the overproduction of reactive oxygen species (ROS) that originates due to the stress generated by the antibiotic present in the plant [37]. This stress arises from the overexcitation of chlorophylls and the disruption of the electron transport chain [38]. Therefore, there is likely a positive relationship between these natural polyphenols in this species and its ability to cope with environmental stress caused by AZM.

On the other hand, it was observed that the tested concentrations of AZM caused a hormetic response in this crop regarding its height or growth. Previous studies observed that enrofloxacin inhibits the formation of nodules in alfalfa roots by up to 91% [33]; therefore, the bacteriostatic capacity of antibiotics, including azithromycin, due to their persistence in soil, could affect the reproduction of beneficial bacteria [39] reducing the nitrogen-fixing capacity. On the other hand, they reveal minimal toxicity of CIP and AZM transmitted through

biosolids to microbial functioning [40]. Other studies suggest that they might also be related to the antibiotic's disruption of nitrogen metabolism in the plant, leading to a significant reduction in nitrate reductase [41, 42]. It is known that nitrate ($NO_3^-$) is a fundamental nitrogen source for plants. For nitrogen metabolism, the enzyme nitrate reductase (NR) converts the nitrate absorbed by plants into nitrite ($NO_2^-$), while the enzyme nitrite reductase converts nitrite into ammonium ($NH_4^+$) [43]; this situation could have occurred inside these plants. Similarly, plant cells avoid ammonium toxicity by quickly transforming it into amino acids through the main conversion pathway involving the sequential action of glutamine synthetase and glutamate synthetase [44]. In that sense, the alteration in nitrogen metabolism leads to an increase in nitrogen content at lower antibiotic concentrations, while it decreases at higher concentrations [45]; this could explain the increase and decrease in stem length observed in the present study. Therefore, it would justify why the species *Zea mays* L. exhibited a hormetic response to the various doses of AZM studied in this study.

## Dry biomass

These results show a significant decrease in the weight of the cob at all three applied doses: 1, 10, and 100 μg/L of AZM, compared to the control group. It is important to note that as the AZM dose increases, there is a progressive decrease in cob weight. In this sense, the group exposed to 100 μg/L of AZM shows the lowest cob weight among all the analyzed groups. These findings highlight the negative influence that AZM has on the development of purple corn cobs.

In corn, there are two types of inflorescences: the tassel containing the male flowers at the plant's apex and another containing the female flowers on the side of the plant [46]. As for the reduction in tassel weight (male flowers), this weight decreased as the applied doses increased. This could be attributed to increased growth hormones such as gibberellins and auxins; according to [47], there was a decrease in the weight of the corn crop inflorescence when gibberellic acid levels increased in younger spikes. Additionally, it has been reported that auxins reduce the fresh weight of the inflorescence in mature spikes. Likewise [46, 48], reported that from 12 days before spiking, the levels of indole-3-acetic acid, zeatin riboside, zeatin, and gibberellic acid in the upper and lower spikes increased dramatically and then decreased rapidly. This suggests a strong relationship between phytohormones and the differentiation of superiority and inferiority in the spikes. In that context, it is possible that AZM, as an abiotic stressor, is contributing to stimulating the increase of these phytohormones in the plant, which could be causing stress in the inflorescence, reducing its dry weight. On the other hand [49], reported weak pigmentation and low concentrations of anthocyanins in the tassel of purple corn, suggesting that AZM overcame the plant's defense system, causing oxidative stress. This could lead to a reduction in the weight of the tassel.

These reveal a significant increase ($P < 0.05$) in stem weight when the plant is exposed to a concentration of 1 μg/L of AZM, compared to the group exposed to 100 μg/L of AZM. This indicates that the applied concentrations of AZM caused a hormetic response regarding stem weight in the species *Zea mays* L. A 14 and 7% increase in stem weight was observed concerning the control group at doses of 1 and 10 μg/L of AZM, respectively. However, at high doses (100 μg/L of AZM), a 15% decrease in stem weight was recorded compared to the control group. These findings highlight the positive and negative influence AZM exerts on stem weight in purple corn. The same phenomenon is observed in leaf weight, as there is a significant difference between the treatment at one and the control group; likewise, once it reaches its peak in leaf weight, it tends to decrease below the control group at the highest dose.

In this study, it was revealed that purple corn could tolerate low concentrations of AZM (1 μg/L of AZM). This is because, at this concentration, an increase in stem weight is observed.

This could be attributed to the action of gibberellins, which induce stem elongation. Along with this effect, there is a decrease in stem thickness and a reduction in leaf size [50]. This explains why a hormetic response is observed regarding stem and leaf weight. On the other hand, this response could also be related to anthocyanins in the stem and leaves [49].

This highlights the central role of anthocyanins as antioxidants, as they protect the plant from stress by eliminating free radicals produced in AZM-induced oxidative stress. However, it appears less effective at high AZM concentrations (100 µg/L of AZM), as a decrease in leaf and stem weight below the control group is observed. On the other hand, in the study of [51], the absorption of AZM by lettuce plants, which were grown in a sand substrate enriched with biological sludge containing concentrations of AZM considered environmentally relevant, was recorded. In addition, another analysis conducted by [24] indicated that when maize undergoes antibiotic-induced stress, it forms smaller vascular bundles, negatively affecting the transport of photosynthetic substances. Therefore, AZM could be mobilized to different parts of the plant, simultaneously impairing the photosynthesis process in maize, resulting in a decrease in leaf and stem weight and, overall, a reduction in the plant's aerial biomass.

### Root length and weight of the root biomass

Table 2 presents the results regarding the root length and weight of the root biomass of purple corn (Zea mays L.).

The results revealed that *Zea mays* L. did not experience significant effects when exposed to AZM at doses of 1, 10, and 100 µg/L during its vegetative period. However, a reduction in root biomass weight was observed by 24 and 16% at concentrations of 1 and 10 µg/L, respectively, compared to the control group. On the other hand, a slight elongation in root length was observed at a concentration of 1 µg/L compared to the control group. Nevertheless, this positive effect disappeared when the concentration was increased to 10 and 100 µg/L of AZM, resulting in a decrease in root length below the levels of the control group. Therefore, our results suggest that purple corn seems resistant to AZM at low doses, but at doses higher than 10 µg/L, it may negatively impact root biomass and length. These findings indicate that the response of *Zea mays* L. to AZM varies depending on the concentration, and higher doses can negatively affect the plant's root system.

In the different treatments with AZM, there is no significant effect on the root length or root biomass weight of *Zea mays* L. This could be attributed to the glutathione S-transferases (GST) detoxification pathway, where the drug is first catalyzed by cytochrome P450, followed by conjugation mediated by glutathione (GSH) in the cytoplasm, and then transported to the vacuolar compartment for elimination outside the cell [52, 23]. Although there was no significant difference between the treatments, a hormetic response has been observed in root length, which may be related to the abscisic acid content in the roots that can serve as a chemical signal

**Table 2. Root length and dry root biomass weight according to AZM concentrations.**

| Concentration of AZM µg/L | Root length (centimeters) | | Dry weight of root biomass (grams) | |
|---|---|---|---|---|
| | X̄ | SD | X̄ | SD |
| 0 | 79.8[a] | ±13.72 | 21.67[a] | ±11.29 |
| 1 | 85.5[a] | ±13.17 | 16.43[a] | ±9.13 |
| 10 | 76.5[a] | ±23.18 | 18.15[a] | ±19.63 |
| 100 | 77.5[a] | ±13.02 | 21.74[a] | ±19.53 |

X̄: Represents the average. DE: Represents the standard deviation (n = 3).

in root development and elongation in response to antibiotic stress. Additionally, there is an increase in the antioxidant system, including superoxide dismutase (SOD), peroxidase (POD), and catalase (CAT); this system can be activated by the overproduction of reactive oxygen species (ROS) [53]. Therefore, the roots are the main entry pathways for antibiotics into plants [54]. The absorption of the antibiotic by plants appears to depend primarily on its solubility and octanol-water partition coefficients (log P o/w); for AZM, its log P o/w is 3.87–4.02, and it also has water solubility $\geq$ 0.5 g /L at 25°C [55, 56]; so, it is within the range (log Kow 1–4) to be easily absorbed by the roots and translocated [57].

## Effect of azithromycin on the content of antioxidant compounds

When the concentration of AZM increased from 0 to 10 µg/L, the content of total polyphenols and monomeric anthocyanins increased two times (Fig 1A and 1B). Although no prior research has specifically explored the impact of antibiotics on polyphenol content in purple corn, recent investigations in diverse plant matrices positively affect the production of these secondary metabolites. For example [11], observed an increase from 1 to 100 µg/L of ciprofloxacin improved by 30% of the total polyphenol content in quinoa grains. This suggests that antibiotics could induce stress conditions within the plant system and promote the production of reactive oxygen species (ROS) or free radicals [58–60]. In response to this stress, plants activate a defense mechanism, synthesizing secondary metabolites (polyphenols) through the shikimic acid pathway [61, 62]. These polyphenols neutralize free radicals by donating electrons (hydrogen) [63].

The total polyphenol content (TPC) is expressed as mg of gallic acid equivalent per gram of dry weight (mg GAE/g dw). The anthocyanin content (AA) is defined as mg of cyanidin-

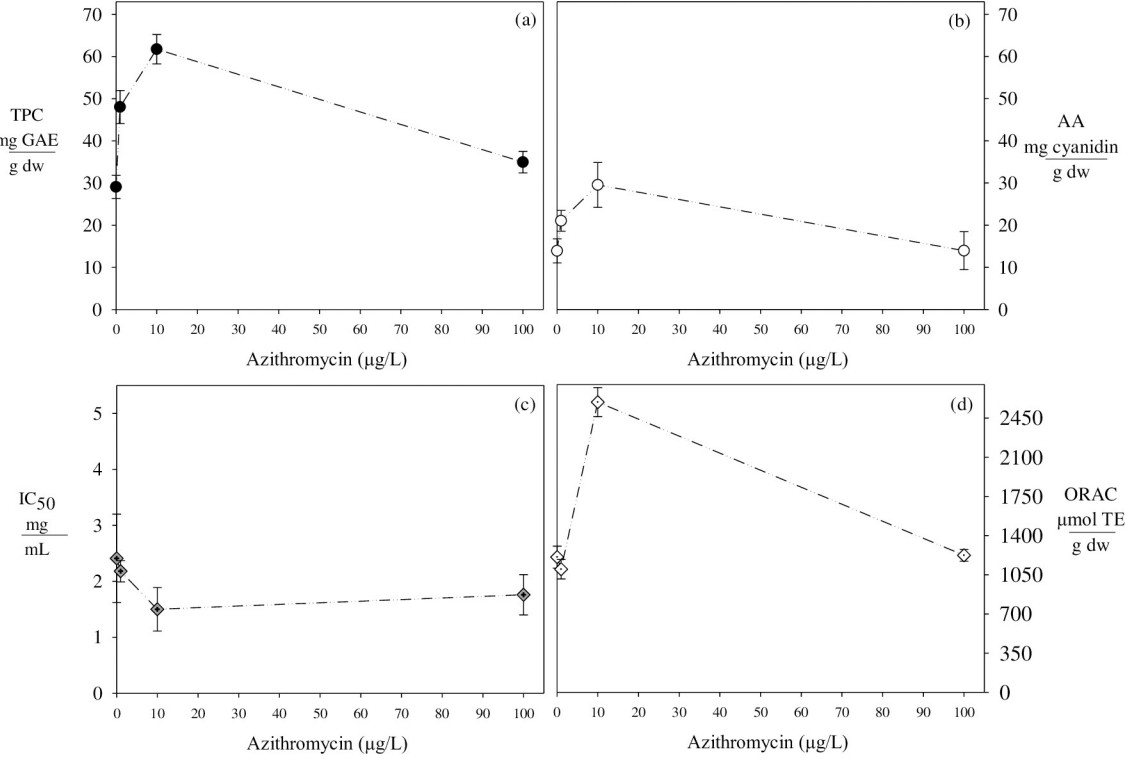

**Fig 1. Behavior of total phenolic compounds, anthocyanins, and antioxidant capacity in the cob of purple corn exposed to azithromycin.**

3-glucoside per gram of dry weight. The $IC_{50}$ value is mg of extract per mL of DPPH solution. The ORAC value is the micromoles of Trolox equivalent per gram of dry weight (μmol TE/g dw). Different letters indicate significant differences for each response variable (p < 0.05).

On the contrary, when the concentration of AZM increased from 10 to 100 μg/L, total polyphenols and monomeric anthocyanins decreased by 44% and 53%, respectively (Fig 1A and 1B). Two distinct inhibition mechanisms are probably happening in the production of secondary metabolites. According to [64], the presence of elevated concentrations of antibiotics can generate an excess of reactive oxygen species (ROS); this surplus of ROS can result in the peroxidation of membrane lipids, causing structural damage that, in turn, diminishes the cells' capacity to synthesize polyphenols [65]. On the other hand, high concentrations of free radicals could inhibit the activity of critical enzymes like phenylalanine ammonia-lyase and coenzyme A (CoA) ligase in the cytoplasm. Consequently, the production of these metabolites could be reduced [12].

The capacity of polyphenols to neutralize free radicals is known as antioxidant capacity. In this context, the DPPH analysis assesses the effectiveness of polyphenols in neutralizing synthetic radicals, while the ORAC analysis evaluates the ability to reduce biological radicals [66]. Thus, conducting both tests to compare the results effectively is advisable. According to our results, ORAC analysis showed the higher the polyphenol content, the higher the antioxidant capacity; conversely, a rise in polyphenol content is associated with a decrease in the IC50 value when it was assessed through the DPPH method (Fig 1C and 1D). In this sense, a high IC50 value indicates that a more significant amount of extract is necessary to inhibit the solution of the DPPH radical.

### Effect of azithromycin on specific families of polyphenols

Similar to the behavior reported for total polyphenols (Table 3), the content of distinct polyphenolic families, including flavonols, phenolic acids, and flavonols, are also affected by the presence of AZM (Table 3). For example, when the AZM concentration increased from 0 to 10 μg/L, the content of flavonols, phenolic acids, and flavonoids increased by 57, 28 and 83%, respectively (Table 3). On the contrary, when the concentration of AZM rose from 10 to 100 μg/L, the content of flavonols, phenolic acids, and flavonoids decreased by 17%, 40%, and 42%, respectively (Table 3). As previously discussed, the presence of low antibiotic

**Table 3. Chemical characterization of specific families of polyphenols.**

| Description | Treatments (control group and concentrations of Azithromycin) | | | | | | | |
|---|---|---|---|---|---|---|---|---|
| | 0 μg/L | | 1 μg/L | | 10 μg/L | | 100 μg/L | |
| | Mean | SD | Mean | SD | Mean | SD | Mean | SD |
| Flavonols (μg/g dw) | | | | | | | | |
| Rutin | 4.68[a] | ±0.83 | 5.87[b] | ±0.12 | 6.71[b] | ±1.09 | 5.75[b] | ±1.07 |
| Quercetin | 0.46[a] | ±0.09 | 0.82[b] | ±0.11 | 1.37[c] | ±0.17 | 0.93[b] | ±0.03 |
| Σ | 5.14 | | 6.69 | | 8.08 | | 6.68 | |
| Phenolic acids (μg/g dw) | | | | | | | | |
| Vanillic | 2.06[a] | ±0.22 | 2.90[b] | ±0.51 | 2.64[b] | ±0.34 | 1.57[c] | ±0.17 |
| Flavanols (μg/g dw) | | | | | | | | |
| Catechin | 0.66[a] | ±0.03 | 0.69[a] | ±0.08 | 0.96[b] | ±0.11 | 0.64[a] | ±0.16 |
| Epicatechin | 2.02[a] | ±0.22 | 4.94[b] | ±0.82 | 3.56[c] | ±0.21 | 1.93[a] | ±0.26 |
| Procyanidin B2 | 0.50[a] | ±0.11 | 0.76[b] | ±0.09 | 0.82[b] | ±0.15 | 0.60[a] | ±0.13 |
| Procyanidin A2 | 0.75[a] | ±0.12 | 1.05[b] | ±0.18 | 1.89[c] | ±0.41 | 1.06[b] | ±0.15 |
| Σ | 3.93 | | 7.44 | | 7.23 | | 4.23 | |

concentrations promotes the generation of metabolites within the plant system. In contrast, high levels of AZM may potentially disrupt the integrity of the cell membrane, subsequently diminishing the synthesis of polyphenols. Finally, using 10 μg/L of AZM allowed for obtaining high concentrations of specific polyphenols with 6.71, 2.64, and 3.56 μg/g dw for quercetin, vanillic acid, and epicatechin (Table 3).

## Conclusion

The presence of AZM in irrigation water at concentrations of 1, 10, and 100 μg/L does not show a statistically significant effect on stem length; however, concerning the weight of the aboveground biomass, the concentration of 1 μg/L exhibits a statistically significant increase compared to the control. On the other hand, starting from a concentration of 10 up to 100 μg/L, this compound reduces the dry weight of the cob and stem by 41 and 15%, respectively. The root biomass's root length and dry weight do not show a statistically significant effect. However, there is a decrease of 24 and 16% in the dry weight of the root biomass at concentrations of 1 and 10 μg/L, respectively. As for the concentration of phenolic compounds and total anthocyanins, they double their concentration at a dose of 10 μg/L, and a decrease is observed at a dose of 100 μg/L by 44 and 53%, respectively. The same pattern is observed in antioxidant capacity; regarding the content of flavonols, phenolic acids, and flavonols, they increase by 57, 28, and 83%, respectively, when the concentration of AZM is from 0 to 10 μg/L. However, with an increase from 10 to 100 μg/L, these compounds decrease by 17, 40, and 42%, respectively. These findings highlight the importance of understanding the impact of pharmaceutical contaminants on plant secondary metabolites and their environmental effects.

## Supporting information

**S1 Table. Behavior of total phenolic compounds, anthocyanins, and antioxidant capacity in the husk of purple corn exposed to azithromycin.**
(PDF)

## Author Contributions

**Conceptualization:** Yoselin Mamani Ramos, Franz Zirena Vilca.

**Formal analysis:** Yoselin Mamani Ramos, Nils Leander Huamán Castilla.

**Investigation:** Yoselin Mamani Ramos, Elvis Jack Colque Ayma, Franz Zirena Vilca.

**Methodology:** Yoselin Mamani Ramos, Nils Leander Huamán Castilla, Elvis Jack Colque Ayma.

**Supervision:** Yoselin Mamani Ramos.

**Writing – original draft:** Yoselin Mamani Ramos, Noemi Mamani Condori, Clara Nely Campos Quiróz.

**Writing – review & editing:** Yoselin Mamani Ramos, Nils Leander Huamán Castilla, Elvis Jack Colque Ayma, Franz Zirena Vilca.

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
