## [Decision Letter · Decision Letter 0]

22 Mar 2024

PONE-D-24-03135Divergent effects of Azithromycin on purple corn (Zea mays L.) cultivation: Impact on biomass and antioxidant compoundsPLOS ONE

Dear Dr. Vilca,

Thank you for submitting your manuscript to PLOS ONE. After careful consideration, we feel that it has merit but does not fully meet PLOS ONE’s publication criteria as it currently stands. Therefore, we invite you to submit a revised version of the manuscript that addresses the points raised during the review process.

We look forward to receiving your revised manuscript.

Kind regards,

Eugenio Llorens

Academic Editor

PLOS ONE

Journal Requirements:

"Resolución de Comisión Organizadora N° 0310–2020—UNAM and Resolución de Comisión Organizadora N° 059–2021—UNAM"

"To the Universidad Nacional de Moquegua for financing this Project (Resolución de Comisión Organizadora N° 727–2022-UNAM, the Resolución de Comisión Organizadora N° 0310–2020—UNAM and the Resolución de Comisión Organizadora N° 059–2021—UNAM). "

"Resolución de Comisión Organizadora N° 0310–2020—UNAM and Resolución de Comisión Organizadora N° 059–2021—UNAM"

"NO authors have competing interests"

6. Please provide a complete Data Availability Statement in the submission form, ensuring you include all necessary access information or a reason for why you are unable to make your data freely accessible. If your research concerns only data provided within your submission, please write "All data are in the manuscript and/or supporting information files" as your Data Availability Statement.

7. PLOS requires an ORCID iD for the corresponding author in Editorial Manager on papers submitted after December 6th, 2016. Please ensure that you have an ORCID iD and that it is validated in Editorial Manager. To do this, go to ‘Update my Information’ (in the upper left-hand corner of the main menu), and click on the Fetch/Validate link next to the ORCID field. This will take you to the ORCID site and allow you to create a new iD or authenticate a pre-existing iD in Editorial Manager. Please see the following video for instructions on linking an ORCID iD to your Editorial Manager account: https://www.youtube.com/watch?v=_xcclfuvtxQ

8. Please include a caption for figure 1.

Reviewers' comments:

Reviewer's Responses to Questions

**Comments to the Author**

1. Is the manuscript technically sound, and do the data support the conclusions?

Reviewer #1: No

Reviewer #2: Yes

Reviewer #3: Yes

2. Has the statistical analysis been performed appropriately and rigorously? 

Reviewer #1: No

Reviewer #2: Yes

Reviewer #3: Yes

3. Have the authors made all data underlying the findings in their manuscript fully available?

Reviewer #1: Yes

Reviewer #2: Yes

Reviewer #3: Yes

4. Is the manuscript presented in an intelligible fashion and written in standard English?

Reviewer #1: No

Reviewer #2: No

Reviewer #3: Yes

5. Review Comments to the Author

Reviewer #1: The manuscript describes the effect of the AZM on the growth of corn and on secondary metabolism. Authors performed experiments with four different concentrations of AZM with four biological replications each.

It seems like the differences observed in this study is difficult to say that it is only caused by the AZM concentration. In such a big size pot experiment, it is hard to be consistent among replications and moreover, such a long term experiment would have more chance to be screwed with unwanted effects. To verify the changes were caused by AZM treatment It should first be confirmed that AZM is transformed into plant tissues. How much is remained in soil, drain water? Is it caused by microbiome in the soil?

The following questions should also be addressed.

L97 Which seeds? Plant materials should be specified. eg, cultivar or inbreds

L108 Be consistent displaying units and spacing between numbers and units, throughout the manuscript, eg, μg or ug in L422, L434, Table 3 treatments, etc.

L125 Which samples? The plants were divided into roots, leaves, and cobs in L118. Did you use whole cob sample?

L234. Table 1. Ear weight was signicantly reduced as the AZM concentration was increased. What about the 100-kernel weight?

Reviewer #2: Recommendation:

1. Please do language check.

2. Please check some reference at no.49 in text , wether it is correct position or not.

3. Introduction : Please explain how AZM will be the residue in water.

4. Line 120: plate height means stem length ?? Please clarify the word and check consistency.

5. Explain how to separate stem and how to measure or give the picture of each part of cob to show your measurement (If possible).

6. Line 131: %/% =??? Please clarify.

7. Line 135: How to be sure that water also removed? , Since you did not freeze dry?

8. Line 136: Clarify how to prepare sample to final volume 10 ml or you add 10 ml of 60% acetone which the final volume of each sample will be the same??

9. Line 141 ; Please rewrite this sentence with reference 26 and 27

10.Line 169: Equation (2) and what are gbh and gbs??

11. Line 179: How to prepare the solution in different concentrations of extracts ?.

12: Line 195-195: Explain the concentration of Trolox and the sample that were used to find IC50.

13. Line 207 : What is 17% ??

14. Line 210: This membrane means the syringe filter ?

15. Line 232 and Table 1: Plant height = stem length?

16. Table 1: Please correct and write the unit for weight, height and concentration of AZM

17. Line 265: How about the effect to nitrogen fixing bacteria? Does AZM can affect? Please explain more .

18. Table2: Add unit and correct the letter which indicated significant difference (a,b)

19. Table 2: Why do 0 and 100 ug of AZM cause similar dry weight of root?

20 Line 390 : Figure 1 has no caption.

21. Line 395: umole ET/g =?? Is it correct?

22. Table 3: Treatments means treatment with ? should give more details

23. References: Check and correct the format.

Reviewer #3: Dear Authors, after a reviewing process I recommend your manuscript to be published at present form.

Review report

Main Question Addressed:

The research investigates the impact of Azithromycin (AZM) residues in irrigation water on the biomass and antioxidant compounds of purple corn. Specifically, it examines how different concentrations of AZM affect various parameters of purple corn growth and antioxidant activity.

Originality and Relevance:

The topic addressed in the manuscript is both original and relevant in the field of agricultural and environmental science. It explores the understudied area of pharmaceutical contamination in agricultural practices, addressing a specific gap in understanding the effects of antibiotics like AZM on plant physiology and secondary metabolites. Given the increasing concern over pharmaceutical pollution in water sources, this research holds significant relevance for agricultural sustainability and food safety.

Contribution to the Subject Area:

Compared to existing literature, this manuscript provides novel insights into the effects of AZM contamination on purple corn. It specifically examines a wide range of concentrations and evaluates their impact on biomass and antioxidant compounds, shedding light on dose-dependent responses. This detailed investigation enhances our understanding of how pharmaceutical contaminants can influence plant physiology and biochemistry.

Methodological Improvements:

While the methodology appears robust, a few improvements could enhance the study's reliability. Firstly, it would be beneficial to include additional controls to account for potential confounding factors, such as variations in soil composition or environmental conditions. Moreover, considering the complexity of plant-soil interactions, incorporating molecular analyses to elucidate underlying mechanisms would strengthen the study's conclusions.

Consistency of Conclusions:

Overall, the conclusions drawn from the evidence presented are consistent with the study's findings and the main question posed. The authors effectively demonstrate how different concentrations of AZM in irrigation water impact purple corn growth parameters and antioxidant compounds. However, providing a more detailed discussion on the implications of these findings for agricultural practices and environmental management would enrich the manuscript.

Appropriateness of References:

The references cited in the seem relevant and appropriate for supporting the research background and contextualizing the findings.

Additional Comments:

The abstract provides clear and concise summaries of the research objectives, methods, results, and conclusions. Including graphical representations, such as figures or tables, summarizing key findings could enhance the abstract's visual appeal and aid in conveying complex data more effectively. Additionally, ensuring clarity in terminology and providing definitions for specialized terms would improve accessibility for readers from diverse backgrounds.

6. PLOS authors have the option to publish the peer review history of their article (what does this mean?). If published, this will include your full peer review and any attached files.

Reviewer #1: No

Reviewer #2: No

Reviewer #3: No

---

## [Author Response · Author response to Decision Letter 0]

5 Jun 2024

Ilo, May 29th, 2024

Mr.

Editor of the Plos One Journal

Present

Subject: Raising of observations

In response to the request for reference and comments made by the referees to the manuscript “Divergent effects of Azithromycin on purple corn (Zea mays L.) cultivation: Impact on biomass and antioxidant compounds” In this regard, at the beginning, we thank the referees for their timely comments on the manuscript, which contributed significantly to its improvement. At the same time, we indicate that the comments have been corrected, also here we respond to the comments made by reviewers:

 Reviewer #1

 The manuscript describes the AZM's effect on corn's growth and secondary metabolism. The authors performed experiments with four different concentrations of AZM with four biological replications each. It seems like the differences observed in this study is difficult to say that it is only caused by the AZM concentration. In such a big size pot experiment, it is hard to be consistent among replications and moreover, such a long term experiment would have more chance to be screwed with unwanted effects. To verify the changes were caused by AZM treatment It should first be confirmed that AZM is transformed into plant tissues. How much is remained in soil, drain water? Is it caused by microbiome in the soil?

Answer:

We greatly appreciate your insightful comments and observations. We acknowledge that it is indeed challenging to categorically state that the differences observed in our study are solely due to the concentration of azithromycin (AZM). However, prior to setting up the experiment, we took several precautions to minimize external variabilities that could potentially affect our results. For instance, we prepared the substrate uniformly and distributed it evenly across the pots to ensure consistency among the experimental units, varying only the concentrations of AZM in the irrigation. The irrigation water used was distilled water to ensure that only the AZM concentrations were present in this water.

Regarding the transformation and presence of AZM in plant tissues, we based our methodology on previous studies that have demonstrated AZM's ability to enter and bioaccumulate in plant tissues due to its physicochemical properties. For example, Sidhu, O’Connor, & Kruse (2019) documented the translocation of azithromycin in plant tissues, which supported our decision not to perform additional quantification of AZM in the tissues in this study. Similarly, Almeida et al. (2021) did not directly quantify azithromycin in exposed algae, focusing instead on biological response parameters such as growth rate and pigment content.

The bioaccumulation of AZM in plants is correlated with its high aromaticity and higher log Dow value, indicating a high potential for bioaccumulation. The octanol-water distribution ratio (Dow) values of organic compounds are often well correlated with absorption in organic tissues (ECETOC, 2013). Consequently, we assumed that the residual concentration of AZM in the soil and drainage water would be minimal, given its preferential absorption by the plant. Additionally, studies such as those by Sidhu, O’Connor, Ogram, et al. (2019) have shown minimal toxicity of AZM at concentrations up to 0.06 mg kg-1 for soil microbial activity, suggesting that in our study, with significantly lower concentrations, the soil microbiome would not have significantly interfered with AZM absorption by purple corn.

To corroborate that the soil microbiome did not affect the absorption of azithromycin, we compared the phenolic compound results in the control (without azithromycin) with the values reported in the literature (Shahidi & Ambigaipalan, 2015). We observed that the concentrations of polyphenols increased with AZM concentrations of 1 and 10 μg L-1, suggesting the translocation of AZM within the plant.

We thank you once again for your valuable observations. While it would be ideal to conduct direct measurements of AZM in plant tissues in future studies, the available evidence and the precautions we have taken support the conclusions expressed in the present study.

Almeida, A. C., Gomes, T., Lomba, J. A. B., & Lillicrap, A. (2021). Specific toxicity of azithromycin to the freshwater microalga Raphidocelis subcapitata. Ecotoxicology and Environmental Safety, 222. https://doi.org/10.1016/j.ecoenv.2021.112553

ECETOC, (European Centre for Ecotoxicology and Toxicology of Chemicals). (2013). ECETOC Technical Report No. 117 - Understanding the Relationship between Extraction Technique and Bioavailability. 117.

Shahidi, F., & Ambigaipalan, P. (2015). Phenolics and polyphenolics in foods, beverages and spices: Antioxidant activity and health effects - A review. Journal of Functional Foods, 18, 820–897. https://doi.org/10.1016/j.jff.2015.06.018

Sidhu, H., O’Connor, G., & Kruse, J. (2019). Plant toxicity and accumulation of biosolids-borne ciprofloxacin and azithromycin. Science of the Total Environment, 648, 1219–1226. https://doi.org/10.1016/j.scitotenv.2018.08.218

Sidhu, H., O’Connor, G., Ogram, A., & Kumar, K. (2019). Bioavailability of biosolids-borne ciprofloxacin and azithromycin to terrestrial organisms: Microbial toxicity and earthworm responses. Science of the Total Environment, 650, 18–26. https://doi.org/10.1016/j.scitotenv.2018.09.004

 The following questions should also be addressed. L97 Which seeds? Plant materials should be specified. eg, cultivar or inbreds

Answer:

Additional information was added in lines L102 to L104 regarding the seeds used in the present study.

 L108 Be consistent in displaying units and spacing between numbers and units throughout the manuscript, eg, μg or ug in L422, L434, Table 3 treatments, etc.

Answer:

The units in Table 3 and the spacing between numbers and units throughout the article's manuscript were modified.

 L125 Which samples? The plants were divided into roots, leaves, and cobs in L118. Did you use whole cob sample?

Answer:

Line L137 specifies the sample. Lines L127 to L131 added more information about using the samples and the division process. Furthermore, regarding the cob, it was specified that only the core was used in lines L131 to L135.

 L234. Table 1. Ear weight was signicantly reduced as the AZM concentration was increased. What about the 100-kernel weight?

Answer:

The term "Ear weight" was changed to "Tassel weight" in Table 1, as grains are not found in the tassel. The grains were not weighed.

 Reviewer #2

 Please do language check.

Answer:

The language and spelling of the entire article were checked.

 Please check some reference at no.49 in text, wether it is correct position or not.

Answer:

Reference No. 49 was checked and the content of the citation was clarified in lines L324 to L327.

 Introduction : Please explain how AZM will be the residue in water.

Answer:

More information was added in lines L70 to L80 about how AZM turns into residue in water.

 Line 120: plate height means stem length ?? Please clarify the word and check consistency.

Answer:

The manuscript's coherence was checked. Additionally, it was clarified that only the term "stem length" will be used throughout the manuscript.

 Explain how to separate stem and how to measure or give the picture of each part of cob to show your measurement (If possible).

Answer:

More information about the measurement process was added in lines L127 to L130. 

 Line 131: %/% =??? Please clarify.

Answer:

The (%/%/) was changed to (v/v) in line L143

 Line 135: How to be sure that water also removed? , Since you did not freeze dry?

Answer:

Additional information about the mechanism for water evaporation was added in lines L147 to L149.

 Line 136: Clarify how to prepare sample to final volume 10 ml or you add 10 ml of 60% acetone which the final volume of each sample will be the same??

Answer:

Lines L150 to L152 detail the process of reconstituting the phenolic compounds in 10 mL of 60% acetone.

 Line 141 ; Please rewrite this sentence with reference 26 and 27

Answer:

The sentence in lines L156 was rewritten.

 Line 169: Equation (2) and what are gbh and gbs??

Answer:

The units in line L184 were modified, and their meanings were added in lines L190 to L191.

TAC ((mg EC3G)/L)= (A×MW ×FD×1000)/eL×(V )/(g fw)×(g fw)/(g dw)

 Line 179: How to prepare the solution in different concentrations of extracts?

Answer:

Additional information was added about the preparation of the solution in different concentrations of extracts in lines L194 to L198.

 Line 195-195: Explain the concentration of Trolox and the sample that were used to find IC50.

Answer:

The concentration of Trolox and the sample used were added in lines L214 to L218.

 Line 207 : What is 17% ??.

Answer:

This refers to the percentage of C18 contained in the extraction cartridge.

 Line 210: This membrane means the syringe filter ?

Answer:

The sentence was modified to explain that it was filtered through a syringe filter with a pore size of 0.22 μm in line L233.

 Line 232 and Table 1: Plant height = stem length?

Answer:

It was clarified that only the term "stem length" will be used throughout the manuscript in Table 1, L257

 Table 1: Please correct and write the unit for weight, height and concentration of AZM

Answer:

The units for weight, height, and AZM concentration were corrected in Table 1.

 Line 265: How about the effect to nitrogen fixing bacteria? Does AZM can affect? Please explain more

Answer:

Additional information about the effect of AZM on nitrogen-fixing bacteria was added in lines L285 to L291.

 Table2: Add unit and correct the letter which indicated significant difference (a,b)

Answer:

The unit was added and the letter indicating a significant difference was corrected in Table 2

 Table 2: Why do 0 and 100 ug of AZM cause similar dry weight of root?

Answer:

When using only one unit, the results are similar. However, when working with 2 decimal places, a variation in the decimal figures is noticeable. For this reason, it was decided to use 2 decimal places in Table 2.

 Line 390 : Figure 1 has no caption

Answer:

The title was added to Figure 1.

 Line 395: umole ET/g =?? Is it correct?

Answer:

In line L412, the units (μmol ET/g dw) were corrected to (μmol TE/g dw).

 Table 3: Treatments means treatment with ? should give more details

Answer:

The treatments are detailed in Table 3.

 References: Check and correct the format.

Answer:

The format was checked and corrected.

 Reviewer #3

Dear Authors, after a reviewing process I recommend your manuscript to be published at present form.

Franz Zirena Vilca

Professor

---

## [Decision Letter · Decision Letter 1]

9 Jul 2024

Divergent effects of Azithromycin on purple corn (Zea mays L.) cultivation: Impact on biomass and antioxidant compounds

PONE-D-24-03135R1

Dear Dr. Vilca,

We’re pleased to inform you that your manuscript has been judged scientifically suitable for publication and will be formally accepted for publication once it meets all outstanding technical requirements.

Kind regards,

Eugenio Llorens

Academic Editor

PLOS ONE

Additional Editor Comments (optional):

Reviewers' comments:

Reviewer's Responses to Questions

**Comments to the Author**

1. If the authors have adequately addressed your comments raised in a previous round of review and you feel that this manuscript is now acceptable for publication, you may indicate that here to bypass the “Comments to the Author” section, enter your conflict of interest statement in the “Confidential to Editor” section, and submit your "Accept" recommendation.

Reviewer #4: All comments have been addressed

Reviewer #5: All comments have been addressed

2. Is the manuscript technically sound, and do the data support the conclusions?

Reviewer #4: Yes

Reviewer #5: Yes

3. Has the statistical analysis been performed appropriately and rigorously? 

Reviewer #4: Yes

Reviewer #5: Yes

4. Have the authors made all data underlying the findings in their manuscript fully available?

Reviewer #4: Yes

Reviewer #5: Yes

5. Is the manuscript presented in an intelligible fashion and written in standard English?

Reviewer #4: Yes

Reviewer #5: Yes

6. Review Comments to the Author

Reviewer #4: This manuscript has undergone a thorough review, and it is evident that the authors have implemented all necessary modifications to address the reviewers’ concerns. The authors' responses to the reviewers are comprehensive and effectively justify their methodological choices and study outcomes. The additional information incorporated into the manuscript is appreciated and enhances its quality, making it suitable for publication.

Reviewer #5: this is a revised version of the article titled Divergent effects of Azithromycin on purple corn (Zea mays L.) cultivation: Impact on biomass and antioxidant compounds, where the authors carrefully addressed the comments

7. PLOS authors have the option to publish the peer review history of their article (what does this mean?). If published, this will include your full peer review and any attached files.

Reviewer #4: No

Reviewer #5: No

---

## [Editor Report · Acceptance letter]

11 Jul 2024

PONE-D-24-03135R1 

PLOS ONE

Dear Dr. Vilca, 

I'm pleased to inform you that your manuscript has been deemed suitable for publication in PLOS ONE. Congratulations! Your manuscript is now being handed over to our production team.

Kind regards, 

on behalf of

Dr. Eugenio Llorens 

Academic Editor

PLOS ONE